# Physicians’ Distress Related to Moral Issues and Mental Health In-Between Two Late Waves of COVID-19 Contagions

**DOI:** 10.3390/ijerph20053989

**Published:** 2023-02-23

**Authors:** Davis Cooper-Bribiesca, Dulce María Rascón-Martínez, José Adan Miguel-Puga, María Karen Juárez-Carreón, Luis Alejandro Sánchez-Hurtado, Tania Colin-Martinez, Juan Carlos Anda-Garay, Eliseo Espinosa-Poblano, Kathrine Jáuregui-Renaud

**Affiliations:** 1Unidad de Investigación Médica en Otoneurología, Instituto Mexicano del Seguro Social, Ciudad de México 06720, Mexico; 2Departamento de Psiquiatría, Hospital de Especialidades del Centro Médico Nacional Siglo XXI del Instituto Mexicano del Seguro Social, Ciudad de México 06720, Mexico; 3Departamento de Anestesiología, Hospital de Especialidades del Centro Médico Nacional Siglo XXI del Instituto Mexicano del Seguro Social, Ciudad de México 06720, Mexico; 4Departamento de Terapia Intensiva, Hospital de Especialidades del Centro Médico Nacional Siglo XXI del Instituto Mexicano del Seguro Social, Ciudad de México 06720, Mexico; 5Departamento de Admisión Continua, Hospital de Especialidades del Centro Médico Nacional Siglo XXI del Instituto Mexicano del Seguro Social, Ciudad de México 06720, Mexico; 6Departamento de Medicina Interna, Hospital de Especialidades del Centro Médico Nacional Siglo XXI del Instituto Mexicano del Seguro Social, Ciudad de México 06720, Mexico; 7Departamento de Inhaloterapia y Neumología, Hospital de Especialidades del Centro Médico Nacional Siglo XXI del Instituto Mexicano del Seguro Social, Ciudad de México 06720, Mexico

**Keywords:** mental health, physicians, moral distress, moral injury, sense of coherence, clinical empathy, resilience, sanitary crisis, COVID-19

## Abstract

In addition to the sanitary constrains implemented due to the pandemic, frontline physicians have faced increased workloads with insufficient resources, and the responsibility to make extraordinary clinical decisions. In 108 physicians who were at the forefront of care of patients with COVID-19 during the first two years of the pandemic, mental health, moral distress, and moral injury were assessed twice, in between two late waves of COVID-19 contagions, according to their adverse psychological reactions, in-hospital experience, sick leave due to COVID-19, quality of sleep, moral sensitivity, clinical empathy, resilience, and sense of coherence. Three months after the wave of contagions, the adverse emotional reactions and moral distress decreased, while moral injury persisted. Moral distress was related to clinical empathy, with influence from burnout and sick leave due to COVID-19, and moral injury was related to the sense of coherence, while recovery from moral distress was related to resilience. The results suggest that measures to prevent physician infection, as well as strengthening resilience and a sense of coherence, may be helpful to prevent persistent mental damage after exposure to a sanitary crisis.

## 1. Introduction

Worldwide, mental reactions to the COVID-19 pandemic have been diverse. A meta-analysis of 55 studies (45 in China; 41 in the general population), including 189,159 participants showed a prevalence of 15.9% (95%CI 13.2–19.1%) for depression, 15.1% (95%CI 12.2–18.5%) for anxiety, 23.8% (95%CI 15.7–34.4%) for insomnia, and 21.9% (95%CI 9.3–43.3%) for posttraumatic stress disorder [1]. A systematic review of 117 studies (65% conducted in Asian countries), including 119,189 health workers, showed a pooled prevalence of 40% for acute stress (*n* = 6949), 30% for anxiety (*n* = 43,751), 24% for depression (*n* = 61,463), 28% for burnout (*n* = 1168), and 13% for post-traumatic stress disorder (*n* = 24,540) [2]. In the United States of America (U.S.A.), the assessment of insomnia in 573 health workers (72% women) showed an increased rate of insomnia disorder related to the pandemic [3].

In any occupation, chronic exposure to work stress can drive burnout [4,5], which is more frequent in physicians than in workers in other fields [6]. In the U.S.A., during the first wave of COVID-19 contagions, among 20,947 health workers, 49% reported burnout and 43% reported work overload [7]. In the same country, a national survey of physicians showed that providing care to patients with COVID-19 was strongly related to burnout; however, compared to a similar survey that was conducted three years earlier, physicians with specialties reported no change in burnout (50.3% in 2017 *versus* 48.6% in 2020) [8]. In the United Kingdom (U.K.), during the second wave of contagions, 851 health workers showed a 3.3-fold increase in the risk of burnout in two domains (emotional exhaustion and depersonalization), compared to 106 non-health workers; whereas patient-facing workers had a 2.7-fold increased risk on emotional exhaustion, compared to non-patient-facing workers (730 patient-facing/113 non-patient-facing workers) [9].

In the context of the extraordinary crisis that was provoked by the pandemic, apart from the burnout and the psychological distress related to healthcare delivery, violations of integrity have been raised. In the patient’s best interest, whenever health resolutions have been made, a context-sensitive approach to moral commitment has been desirable; as well as the understanding of patient suffering, the capacity to communicate this understanding, and the intention to help (clinical empathy) [10]. However, for any specific clinical situation, individual values/aptitudes may have entailed variable levels of moral distress [11].

Moral distress arises under circumstances in which “one knows the right thing to do, but institutional constraints make it nearly impossible to pursue the right course of action” [12]. In healthcare, this concept is clarified by the notion that “in morally conflicting situations, the commitment to professional values and experiencing meaning in all patient care is threatened” [13]; encompassing risk factors related to the patient, the health workers, and the institution/system [14]. In the U.S.A., 52.7% to 87.8% of 2579 health workers providing care to patients with COVID-19 (42% nurses), reported initial moral distress related to the fear of infecting others, the negative impact on family, and concerns related to work [15]. In the same country, in 307 nurses (41% in acute care), increased moral distress was related to patient workload and to the personal protective equipment workarounds [16]; while during the first wave of COVID-19 contagions, a follow-up of 378 health workers showed that moral distress predicted individual mental health strain and burnout, even though it was not a predictor of maladaptive coping [17]. Contrariwise, in the Netherlands, in 345 nurses and 103 supporting staff, moral distress was significantly lower compared to one year before COVID-19, since the ethical climate was rated positively regarding the culture of mutual respect, ethical awareness, and support [18].

Sustained moral distress may evolve into moral injury, which results from acts of perpetration or omission or the experience of betrayal from others, which violate the individual moral code and impair the capacity for trust [19]. Although moral injury has been largely assessed in military populations, knowledge about civilians exposed to trauma is yet limited (for review see [20]). Studies in health workers have shown emotional exhaustion and psychopathology related to moral injury [21], which have been more evident during the COVID-19 pandemic [22,23,24]. In the U.S.A., just prior to the pandemic, an online cross-sectional study of 181 health workers (70.7% physicians) showed moral injury symptoms causing moderate to extreme problems with family, social, and occupational functioning [21]. In the same country, during the first wave of contagions (March to July 2020), an email-based, longitudinal survey of 96 frontline health workers (62% physicians) showed that after the first three months of the pandemic moral distress decreased, while moral injury remained stable [22]. Moreover, in the U.S.A., at the end of the year 2020, an online study following 267 health workers (15% physicians) for 90 days, showed a high level of moral injury related to psychopathology, especially among those reporting COVID-19 exposure [23]. In Israel, after the third wave of contagions, an online cross-sectional study showed that 31.8% of 296 health workers (36.3% physicians) reported moral injury perpetrated by them, 49.3% reported moral injury perpetrated by others, and 62.2% reported distress related to betrayal [24].

Contrariwise, resilience entails the capacity to recover from extremes of trauma and stress [25]; furthermore, the sense of coherence enables coping in the presence of external stressors by viewing one’s life as comprehensible, manageable, and meaningful [26,27]. In Italy, in 267 health workers (41% physicians) the risk of burnout was inversely related to resilience [28]; in the same country, in 233 health workers (*circa* 10% physicians), the use of neurotic and immature psychological defenses was associated with greater stress and lower resilience [29]. In Spain, 80.6% of 1459 health workers (16.0% physicians) reported psychological distress that was inversely related to their sense of coherence [30].

During the first waves of the pandemic, health workers were continuously facing difficult ethical decisions, according to their individual values, and in the context of their healthcare system. Although literature is emerging on the moral distress and moral injury of health workers during the pandemic, most of the studies have been web-based, in samples including diverse health occupations, and in a variety of working conditions [16,17,18,22,23,24,28,29,30,31].

The pandemic is an unprecedented crisis that heightened distress, particularly among frontline professionals immersed in the care of hospitalized patients with COVID-19. Since evidence supports that the negative outcomes related to moral issues may have been contingent on the relationship between individual responses and the working environment [31,32], the assessment of professionals working within the same context may contribute to understanding the diversity of individual responses to the crisis. Face-to-face studies by occupation, and within the working environment, may contribute to designing more specific strategies to tackle and prevent long-lasting mental harm after a sanitary emergency, especially in those who hold onto providing acute care.

This study was designed to assess distress related to mental health (anxiety/depression/burnout/stress reactions) and moral issues (distress/injury), in physicians who were at the forefront of care of patients with COVID-19 during the first two years of the pandemic, working in the same institutional context. It was performed in between two late waves of COVID-19 contagions and taking into account individual experience and mental resources by assessing the in-hospital experience, the history of sick leave due to COVID-19, the quality of sleep, as well as resilience, clinical empathy, sense of coherence, and moral sensitivity.

## 2. Materials and Methods

The study protocol was approved by the institutional Research and Ethic Committees (R2022-3602-023), and written informed consent was obtained from all the participants.

### 2.1. Setting

The research was performed in a teaching hospital that was reconfigured to address the surge of patients with COVID-19 during the first two waves of the pandemic. Afterwards, the hospital was reorganized to continue providing specialized care; although, a restricted area was organized to isolate sporadic patients who were diagnosed with COVID-19.

### 2.2. Participants

Candidates to participate were eligible if, at the moment of starting the study, they were physicians working in the Departments of Emergency Care, Internal Medicine, Respiratory Support, Anesthesia, and Intensive Care, who had provided medical care to patients with COVID-19 during any of the first four waves of contagions (March 2020 to March 2022). At the time of inclusion in the study, all the participants were fully vaccinated against SARS-CoV2.

In a single center, without sample size calculation, after an open invitation, among 139 medical specialists/residents who showed interest in participating in the study, 109 (78.4%) accepted to participate. However, one of them did not complete all the responses to the instruments. Then, 108 participants completed the study protocol (Appendix A). Among them, 11 staff specialists and 2 medical residents (12%) had participated in a previous research protocol, which was designed to assess stress in frontline health workers during the first two waves of the pandemic.

The 108 physicians who participated in the study were 62 women and 46 men, aged 26 to 45 years (mean age 30.4 ± standard deviation 4.9 years), including:23 medical specialists holding a staff position, who were aged 29 to 45 years (37.3 ± 4.8 years), with a mean in-hospital experience of 112 ± 58 months;85 medical residents aged 26 to 44 years (28.6 ± 3.0 years), with a mean in-hospital experience of 30 ± 21 months. According to their experience during the pandemic, they were categorized into two groups: 58 residents who have been at the forefront of the medical care of patients with COVID-19 since the first wave of contagions (before vaccination started), who were aged 26 to 44 years (29.0 ± 3.3 years), and had a mean in-hospital experience of 38.7 ± 21.4 months; and 27 residents who participated in patient care since the third wave of contagions (after vaccination), in a variety of institutional hospitals, who were aged 26 to 33 years (27.2 ± 1.75 years), and had a mean in-hospital experience of 12.6 ± 0.9 months.

At the time of inclusion in the study, most of the participants were single (83%), 12 (11%) participants had children, and one (1%) was pregnant. The majority of them (81%) reported practicing a religion (74% Catholic, 7% Christian) or “having faith” (5%), and 15 (14%) reported no faith. Tobacco consumption was reported by 8 (7%) participants and alcohol consumption was reported by 72 (66%) participants, but none of them reported consumption of >8 drinks/week. Eighteen (16%) of the participants reported medical prescription due to comorbidities, the most frequent were anxiety/depression (9%) and cardiovascular disease (3%).

During the pandemic, four (4%) participants required medical attention due to general disease (three musculoskeletal/one urological). However, 76 (70%) participants reported previous sick leave due to COVID-19 (confirmed by PCR in 68) but none of them required hospitalization. The frequency of COVID-19 sick leave was similar in the three groups of participants (*p* > 0.05), showing equivalent exposure to the disease; it was 61% (*n* = 14) in staff specialists, 74% (*n* = 43) in the residents who provided health care since the first wave of contagions, and 70% (*n* = 19) in the residents who provided health care since the third wave of contagions.

### 2.3. Procedures

The study protocol included two evaluations:The first evaluation was performed at the end of the fourth wave of contagions (April–May 2022), by the beginning of the residents’ academic year.The second evaluation was performed three months later, at the beginning of the fifth wave of contagions (July–August 2022).

After each of the two evaluations, an independent psychiatrist revised all the responses to the instruments, using the accepted criteria for each instrument and according to DSM-5 criteria [33]; when required, psychiatric recommendations were provided.

In the first evaluation, preliminary assessments were performed on demographics, in-hospital experience, medical history (including sick leave due to COVID-19), as well as screening on resilience (Resilience Scale) [34]; trait anxiety/depression (Hospital Anxiety and Depression Scale, HADS) [35]; and, in the two evaluations, screening on acute and chronic stress reactions by the revised Stanford Acute Stress Questionnaire [36,37] and the Posttraumatic Stress Disorder Symptom Severity Scale-Revised [38].

In each of the two evaluations, with three months in between, the following assessments were performed:Sleep quality by the Pittsburg Sleep Quality Index [39];State Anxiety by the short version of the State-Trait Anxiety Inventory (STAI) [40];Burnout by the short version of the Burnout Measure [5];Dissociation symptoms by the Dissociative Experiences Scale [41] and the Depersonalization/Derealization Inventory [42];Sense of coherence by the short version of the Orientation to Life Questionnaire [27],Clinical empathy by the Jefferson Scale of Empathy [10];Moral sensitivity, moral distress, and moral injury by the Moral Sensitivity Questionnaire [43]; the seven top-ranking situations in intensive health care [44] of the revised Moral Distress Scale [45], with four additional situations reflecting COVID-19 context [16]; and the Moral Injury Event Scale [46].

### 2.4. Instruments

-The Hospital Anxiety and Depression Scale [35] comprises 14 items, 7 for anxiety and 7 for depression, which are rated on a 4-point scale (0 to 4). A total score is calculated by summing the ratings for all the items, and by summing the ratings for the seven items of each subscale to yield two separate sub-scores, which range from 0 to 21, with a cut-off of ≥8 [47]; the internal consistency has shown Cronbach alpha coefficients from 0.67 to 0.93 [48]. The Spanish version has shown a Cronbach alpha of 0.86, for both anxiety and depression, and test–retest reliability with coefficients >0.85 [49].-The short version of the Burnout Measure [5] comprises 10 items rated on a 7-point scale (1 to 7). A total score is calculated by the sum of all the components. The internal consistency coefficients range from 0.85 to 0.92 [5].-The Moral sensitivity questionnaire [43] comprises 9 items worded as assumptions related to patient care, which are rated on a 6-point scale (1 to 6). These nine items are grouped into three factors: Moral strength (3 items); Sense of moral burden (4 items); and Moral responsibility (2 items). The ratings of the items are added, and a higher total score indicates higher sensitivity. It has shown a Cronbach’s alpha coefficient >0.7 [50].-The seven top ranking situations in intensive health care [44] of the revised Moral Distress Scale [45], with four additional situations reflecting COVID-19 context [16]: “caring for patients who: must experience hospitalization without family presence; die during a hospitalization without family and/or clergy present; present transmission risk to your family/household and …being assigned/floated to a new unit, requiring unfamiliar skills or procedures”. Each situation was rated for both frequency and intensity of disturbance, using a 4-point Likert scale (from 0 to 4). The total score for each instrument was calculated by summing the products of the two scores for each item; in the original study, a psychometric evaluation showed two factors (traditional moral distress and COVID-19-specific moral distress), and a Cronbach’s alpha ≥ 0.74 [16].-The Moral injury Event Scale comprises 9 items that are rated using a 6-point Likert-type scale (1 = strongly disagree, 6 = strongly agree) [46]. The psychometric evaluation showed favorable internal validity, temporal stability, initial discriminant, and concurrent validity; the factor analysis indicated a two-factor solution (perceived transgressions and perceived betrayal), while the internal consistency estimate was 0.90 [46]. The scale has been used in health workers in the context of the COVID-19 pandemic [23,31], with an internal consistency of α = 0.87 [23]. For the current study, the original instructions were adapted to ask the respondent to recall his/her individual experiences during the COVID-19 pandemic, which were revised by five psychiatrists until absolute agreement.-The Pittsburg Sleep Quality Index [39] comprises 19 items, with 7 components on subjective sleep quality, sleep latency, sleep duration, habitual sleep efficiency, sleep disturbances, use of sleeping medication, and daytime dysfunction. A total score is calculated by the sum of all the components, and a cut-off >5 is used to distinguish good from poor sleepers. Cronbach’s alpha coefficient has varied from 0.70 to 0.83 [51]. The Spanish version has shown an alpha coefficient of 0.80 and a Spearman correlation coefficient of 0.77 [52].-The Jefferson Scale of Empathy [10] is used to measure empathy in health professionals. The instrument was developed for administration to practicing health professionals. It comprises 20 items that include three factors: perspective-taking, compassionate care, and walking in patient’s shoes. Items are rated using a 7-point Likert-type scale (1 = strongly disagree to 7 = strongly agree), half of them are positively worded, and the other half are negatively worded (reverse scored). A total score is calculated by summing the ratings for all the items (range 7–140). It has been translated into 56 languages, with adequate construct validity, criterion-related validity, and internal consistency reliability (Cronbach’s alpha range 0.80 to 0.85) [10].-The Resilience Scale by Connor and Davidson (2003) [34] comprises 25 items, where each one is rated on a 4-point scale (0 to 4). A total score is calculated by summing the ratings for all the items (range 0 to 100). It has good ratings on content validity, internal consistency, validity, reproducibility, agreement, and reliability [53]. The Spanish version has shown a Cronbach alpha coefficient of 0.86 [54].-The short version of the Orientation to Life Questionnaire [27] comprises 13 items to measure comprehensibility, manageability, and meaningfulness. Comprehensibility refers to the extent to which one perceives internal and external stimuli as rationally understandable; manageability is defined as the resources at one’s disposal that can be used to meet the requirements of the stimuli; and meaningfulness, refers to the feeling that life has an emotional meaning. The responses are provided using a semantic scale of 1 to 7 points (extreme feelings). A total score is calculated by summing the ratings for all the items (range 13 to 91 points). Antonovsky intended the use of a single total score and not of component scores [26]. The instrument has been used in 49 languages in 48 countries. The scale has shown internal consistency with a Cronbach’s alpha of 0.70 to 0.92 [55].-The short version of the State-Trait Anxiety Inventory [40] comprises 6 items coded on a 4-point scale (from 0 to 3). A total score is calculated by the sum of all the components (range 0 to 18). It has shown similar scores to those produced with the full form across subject groups [40]. The internal consistency when administered repeatedly was <0.80 [56].-The Dissociative Experiences Scale [41] comprises 28 items including: disturbances in memory, identity, and cognition, and feelings of derealization, depersonalization, absorption, and imaginative involvement. Scores on each item may range from 0% to 100% by multiples of ten (10%, 20%, 30%, etc.); a total score is calculated as an average of the individual scores (range from 0% to 100%); the low normal range has been considered by a cut-off point of 8. The Spanish version has shown a Cronbach alpha of 0.96 [57].-The Depersonalization/Derealization Inventory by Cox & Swinson (2002) [42] comprises 28 items rated on a 4-point scale (0 to 4). It was designed for the assessment of clinical anxiety states, rather than in the context of dissociative disorders. A total score is obtained by the sum of the individual scores (range 0 to 112), with an internal consistency coefficient of 0.95 [40]. The Spanish version has been used in the general population and in patients with sensory deficits [58].-The Stanford Acute Stress Reaction Questionnaire II (SASRQ-II) [36,37]. At first, the respondent is asked to describe the stressful event and how much disturbance it caused. The second part of the instrument contains 30 statements with answers on a 6-point scale (from 0 = not experienced to 5 = very often experienced). The third part of the instrument includes four questions about the duration of symptoms, degree of functional impairment, and whether the respondent suffers from any mental illness. The range of scores is 0–150. In nurses, it has shown an alpha Cronbach of 0.97 [37].-The Posttraumatic Stress Disorder Symptom Severity Scale-Revised by Echeburúa (2016) [36] is a 21-item structured interview based on DSM-5 criteria. It has shown high internal consistency with a Cronbach alpha of 0.91), as well as good discriminant and concurrent validity; and a diagnostic efficacy of 82.48%, using a cut-off point of 20 [38].

### 2.5. Statistical Analysis

The sample characteristics are described via descriptive statistics. Statistical analysis was performed after assessing data distribution by Kolmogorov Smirnov test; accordingly, data is described using either mean and standard deviation or median and quartiles 1 and 3.

Bivariate analyses were performed using the Wilcoxon paired test, the McNemar chi-square test, and the Kruskal–Wallis analysis of variance, with Holm–Bonferroni correction.

Correlations were explored according to data distribution, using either Pearson coefficient or Goodman–Kruskal Gamma coefficient.

Multivariate analysis of covariance with repeated measures was used to analyze the results on the moral distress instruments (normal distribution), and multivariable regression analyses were used to analyze the results on the Moral Injury Event Scale (gamma distribution), by generalized linear models with log-link function, Wald test and type 3 likelihood ratios; both, multivariate and multivariable analyses were performed including age, sex, and the cofactors included in the aim of the study. The analyses were performed setting the significance level at 0.05.

## 3. Results

### 3.1. Psychological Screening

The frequency of adverse emotional reactions is shown in Figure 1. Compared to the second evaluation, it was higher in the first evaluation (McNemar test, X^2^ = 19.780, *p* < 0.0001):
In the first evaluation, which was performed at the end of the fourth wave of contagions, 44 (41%) participants were experiencing adverse emotional reactions, comprising:-Anxiety/depression conditions in 15 (14%) participants (2 staff specialist/13 residents);-Emotional responses to loss in four (4%) residents;-Burnout syndrome in one (1%) resident;-Adaptation responses to stress in 22 (20%) participants (4 staff specialist/18 residents);-Acute stress reaction in one (1%) staff specialist;-Posttraumatic stress disorder in one (1%) staff specialist.
In the second evaluation three months later, 10 (9%) participants were experiencing adverse emotional reactions, comprising:-Anxiety/depression conditions in 5 (4%) participants (1 staff specialist/4 residents), including four with persistent depression/anxiety and one with persistent anxiety;-Emotional responses to loss in one (1%) resident;-Persistent burnout syndrome in one (1%) resident;-Adaptation responses to stress in two (2%) residents;-Posttraumatic stress disorder in resolution in one (1%) staff specialist.


### 3.2. Bivariate Analysis

A comparison between the first and second evaluations on the total score of each instrument showed a decrease in depersonalization/derealization symptoms and in moral distress (Table 1); while the quality of sleep showed no change, in the first evaluation it was bad in 77% (*n* = 84) of the participants, and in the second evaluation it was bad in 73% (*n* = 79) of the participants; similarly, the Moral Sensitivity Questionnaire showed almost the same scores in the two evaluations, in both men and women (Table 1).

Comparisons among the three groups of participants (according to their in-hospital experience) on each instrument are described in Table 2. A detailed analysis of moral distress showed that, among the items of moral distress related to the COVID-19 situations, the two groups of residents scored higher than the staff specialists on “Being assigned/allocated to a new unit, requiring unfamiliar skills or procedures”, just in the first evaluation (Table 3). This result is consistent with the experience of each group. However, no differences were observed in the total score (Table 2) or on the sub-scores of the Moral Injury Event Scale (Figure 2).

An assessment of the linear relationship between the scores of two evaluations showed significant correlations for all the instrument pairs, which was strong (>0.5) [60] for the Pittsburg Sleep Quality Index, the State-Trait Anxiety Inventory, the Orientation to Life Questionnaire, the Burnout Measure and the Depersonalization/Derealization Inventory (Pearson’s r (108) from 0.50 to 0.60, *p* < 0.0001); it was moderate (0.3 to <0.5) [60] for the Jefferson Scale of Empathy, the Dissociative Experiences Scale, the moral distress related to COVID-19 situations and the Moral Injury Event Scale (Pearson’s r (108) from 0.34 to 0.44, *p* < 0.0001); and it was weak (<0.3) [60] for the Moral Sensitivity Questionnaire and for the Moral Distress Scale-R items (Pearson’s r (108) from 0.22 to 0.27, *p* < 0.02).

An assessment of the linear correlation between the score on the Resilience scale, which was administered only in the first evaluation, and all the other instruments showed a consistent linear correlation of resilience with the score on the Orientation to Life Questionnaire, the correlation was strong in the first evaluation (Pearson’s’ r (108) = 0.55, *p* < 0.0001), and it was moderate in the second evaluation (Pearson’s r (108) = 0.36, *p* < 0.0001). The score on the Resilience scale also showed a moderate inverse correlation with the instruments assessing anxiety (HADS and STAI) (Pearson’s r (108) from −0.36 to −0.49, *p* < 0.0001), while its correlation was weak to moderate with the Moral Sensitivity Questionnaire (Pearson’s r (108) = 0.25 and 0.30, respectively, *p* < 0.01) and the Burnout Measure (Pearson’s r (108) = 0.28 and 0.36, respectively, *p* < 0.0001).

An assessment of the linear correlation between the score of the Moral Distress Scale-R items and the scores on the other instruments administered in each of the two evaluations showed that, in the first evaluation, the correlation was moderate with the scores on the Burnout Measure, the Dissociative Experiences Scale and the Moral Injury Event Scale (Pearson’s r (108) = 0.30 to 0.48, *p* ≤ 0.001), but it was weak with the scores on the Orientation to Life Questionnaire, the HADS, the STAI, and the Depersonalization/Derealization Inventory (Pearson’s r (108) = 0.20 to 0.27, *p* ≤ 0.03); while in the second evaluation, the correlation was strong with the moral distress related to COVID-19 situations (Pearson’s r (108) = 0.69, *p* < 0.0001), and moderate with the STAI, the Burnout Measure and the Moral Injury Event Scale (Pearson’s r (108) = 0.30 to 0.38, *p* ≤ 0.001).

An assessment of the linear correlation between the score of the moral distress related to COVID-19 situations with the other instruments, in the two evaluations, consistently showed a strong correlation with the Moral Distress Scale-R items (Pearson’s r (108) = 0.63 and 0.69, respectively, *p* < 0.0001), as well as moderate correlation with the Moral Injury Event Scale (Pearson’s r (108) = 0.45 and 0.48, respectively, *p* ≤ 0.002), but weak correlation with the Resilience Scale, the HADS and the Depersonalization/Derealization Inventory (Pearson’s r (108) = 0.25–0.27, *p* < 0.01).

An assessment of the non-Gaussian correlation (Goodman–Kruskal Gamma coefficient) between the total score and sub-scores on the Moral Injury Event Scale and the scores on the moral distress instruments showed consistent weak to moderate correlation between evaluations and within the same evaluation (G (108) = 0.26 to 0.36, *p* ≤ 0.002) (Table 4).

Contrariwise, correlation assessment between the score and sub-scores on the Moral Sensitivity Questionnaire and the moral distress and moral injury instruments showed no evident correlation. However, the paired total score and sub-scores obtained in the two evaluations on the Moral Sensitivity Questionnaire were correlated between them, showing G (108) = 0.53,(*p* < 0.0001) for the sub-score on moral burden, and G (108) = 0.23 to 0.26 (*p* < 0.001) for the other sub-scores and total score.

### 3.3. Multivariate Analysis on the Moral Distress Instruments

The multivariate analysis of covariance on the score of the Moral Distress Scale-R items showed that, in the two evaluations, this score was consistently related to the score on the Jefferson Scale of Empathy, which had no influence on the score change at follow-up (Table 5). Nevertheless, in the first evaluation, this score was also related to the score on burnout and the history of sick leave due to COVID-19, which also had an influence on the score change at follow-up (Figure 3); while in the second evaluation, the score on the Moral Distress Scale-R items was related to the score on the Resilience Scale, which had an influence on the score change at follow-up (Table 5).

A group effect was observed in the first evaluation when residents with less experience in having an adverse psychological reaction showed the highest scores on moral distress (F = 4.88, *p* = 0.009). However, no relationships were observed with the age or sex of the participants, the Orientation to Life Questionnaire, the HADS, the STAI, the Pittsburgh Sleep Index, or the scores on the Moral Sensitivity Questionnaire, which were homogeneous.

The multivariate analysis of covariance on the score of the moral distress related to COVID-19 situations showed inconsistent results within each evaluation, but significant differences between evaluations (F = 5.149, *p* = 0.025), with varied change among the three groups of participants; residents who were at the forefront since the first wave of contagions reported persistent distress, while those who were there since the third wave of contagions, as well as the staff specialists, reported less distress (Table 2).

The multivariable analysis on the scores and the sub-scores of the Moral Injury Event Scale showed that (Table 6), in the first evaluation, the total score was related to the scores on the Orientation to Life Questionnaire, the STAI, the Dissociative Experiences Scale, the Moral Distress Scale-R items and the moral distress related to COVID-19 situations. The majority of these relationships persisted in the second evaluation, except for the Orientation to Life Questionnaire. However, the group experience showed no contribution by itself, just when the history of sick leave due to COVID-19 was considered since the less experienced residents with a history of sick leave due to COVID-19 showed the highest scores among the participants.

Overall, the results on the total score of the Moral Injury Event Scale were mainly subtended by the sub-score on Betrayal, with similar results to those obtained on the analysis of the total score (Table 6); while, in the first evaluation, the sub-score on Transgression was related just to the score on moral distress related to COVID-19 situations; and, in the second evaluation, it was also related to the history of sick leave due to COVID-19, the experience group and the score on the Moral Distress Scale-R items (Table 6). No other relationships were observed with the age or sex of the participants, or to adverse psychological reactions, or the scores on the Moral Sensitivity Questionnaire, the Jefferson Scale of Empathy, or the Pittsburgh Sleep Index.

## 4. Discussion

After three months from the fourth wave of contagions, recovery from adverse emotional reactions was evident in the majority of cases, with a general decrease in the depersonalization/derealization symptoms and moral distress, while moral injury persisted. Consistent relationships were observed between moral distress and clinical empathy, and between moral injury and the sense of coherence; however, any relation to moral sensitivity could not be addressed, mainly due to the homogeneity of the responses to the assessment.

Although burnout and sick leave due to COVID-19 were related to moral distress, these relations were not sustained, since burnout persisted while moral distress showed a decrease. Additionally, a group effect was observed on the history of sick leave due to COVID-19, participants who were on sick leave due to COVID-19 showed a high level of distress that decreased at follow-up, while those who reported no sick leave had almost no distress change. During follow-up, resilience was related to distress recovery, showing its contribution to emotional recovery after the epidemic crisis.

In addition to the sanitary constraints implemented for the general population due to the pandemic, health workers have faced increased workloads, with insufficient resources, uncertain clinical protocols and treatments, and the responsibility to take extraordinary clinical decisions [61]. Then, a variety of stressors threatened individual safety and values/integrity generating moral distress and moral injury. The results of this study are consistent with a previous report showing that, during the first wave of contagions in the U.S.A., frontline health workers reported that moral distress decreased after three months, while moral injury persisted [22]. Similar results were obtained in this study, ratings on moral distress decreased over time, while ratings on moral injury remained the same, reflecting psychological harm [62]. The two studies support that, after sustained morally distressing situations, their effect increase over time [14].

Worldwide, the scarcity of human resources to take care of the surge of patients required that low-experienced residents worked in highly demanding settings [63]. In this study, residents reported moral distress related to their work in a clinical area requiring unfamiliar skills or procedures, which decreased at follow-up, when they acquired more experience; consistently, the less experienced residents showed a larger decrease than the more experienced residents. Even though, during the first evaluation, those with a history of sick leave due to COVID-19 were particularly distressed and reported higher ratings on moral injury in the two evaluations. However, since moral injury was reported similarly by the three groups of participants in the two evaluations, this study cannot help to elucidate the effects on moral injury related to the time of in-hospital experience.

In this study, moral injury was particularly related to moral distress in situations related to COVID-19. In the U.S.A., by the last trimester of the year 2020, an online follow-up study showed that circa half of 350 health workers (e.g., nurses, physicians, emergency medical technicians) reported perceived transgression by others and perceived betrayal [23].

Prior to the COVID-19 pandemic, no correlation between clinical empathy and moral distress was observed in physicians, nurses, and residents attending an intensive care meeting [64], and evidence on trainees suggested that it may decrease during medical training [65] contributing to detachment from patients [66]. In this study, we observed an association between moral distress and clinical empathy. Under the extreme demands of COVID-19 health care, ethical dilemmas were common, including the uncertainty of treatment and prognosis as a source of confusion and potential conflict. Additionally, the report of angst was also related to their own experience of the disease, which may have influenced their empathy.

The empathy ratings of physicians with diverse hospital experience were consistently related to moral distress, in two evaluations with three months in between, while burnout showed influence just in the first evaluation, and both moral distress and depersonalization/derealization symptoms decreased in between the two evaluations. The results support a multifactorial perspective on the influence of clinical empathy on moral distress. The findings suggest that, in frontline physicians, a disproportionately high risk of exposure to adverse events may require a close evaluation of mental health outcomes related to trauma.

Nonetheless, in this study, the moral injury was related to the sense of coherence, which refers to people’s ability to perceive a stressful situation as understandable, manageable, and meaningful, allowing them to deal with it when providing healthcare [67]. This finding is consistent with the evidence showing a relation between the sense of coherence and perceived health (in general) and mental health (in particular) [68], as well as with previous evidence documenting its relation with mental health in the clinical setting [69]. In agreement with these studies, during the pandemic, the sense of coherence has been related to psychological distress, and health workers with psychological distress have reported a lower sense of coherence than those with less psychological distress [30]. In addition, we observed that the scores on moral injury were related to state anxiety and dissociation, but not to posttraumatic stress disorder. This result is consistent with the evidence that, in military personnel, moral injury and symptoms of posttraumatic stress disorder may have dissociable neural underpinnings [70].

The stability of the outcomes between evaluations supports that the findings were not just related to time. However, to interpret the results, several limitations have to be considered. The low variability in the responses to the Moral Sensitivity Questionnaire, within each evaluation and between evaluations, did not allow us to explore its potential influence on the results. Since the quality of sleep was bad in the two evaluations in the majority of participants of the study, it is hard to elucidate its influence on the results; although, it is already known that sleep disturbances are frequent among clinical workers, even before the pandemic [71]. The lack of influence from sex or age on the results could be attributed to the characteristics of the sample, which comprised only physicians of similar age, who were working under similar circumstances. Studies in health workers showing sex differences have been performed in samples comprising mainly nurses, showing that female nurses are more likely to experience greater psychological burden than male nurses [72]. In addition, the limited sample size and the narrow age range of the participants precluded adequate evaluation of the influence of age on the responses; although, the in-hospital experience was accounted as a cofactor. Accordingly, before any generalization is made, further studies are required to replicate and amplify the observations of the current study.

## 5. Conclusions

In physicians at the forefront of an epidemic crisis, such as the COVID-19 pandemic, individual resources and their own experience of the disease may have an influence on both moral distress and moral injury. However, moral distress may be remitted according to individual resources, particularly resilience, while moral injury may persist for at least three months. The results suggest that regular psychological assessment of physicians who are at the forefront of health care may allow identifying their coping resources and psychological symptoms that could be relevant to face a sanitary crisis; when required, strengthening of the individual reserves and organizational strategies may contribute to preventing persistent mental damage after exposure to a morally injurious event.

## Figures and Tables

**Figure 1 ijerph-20-03989-f001:**
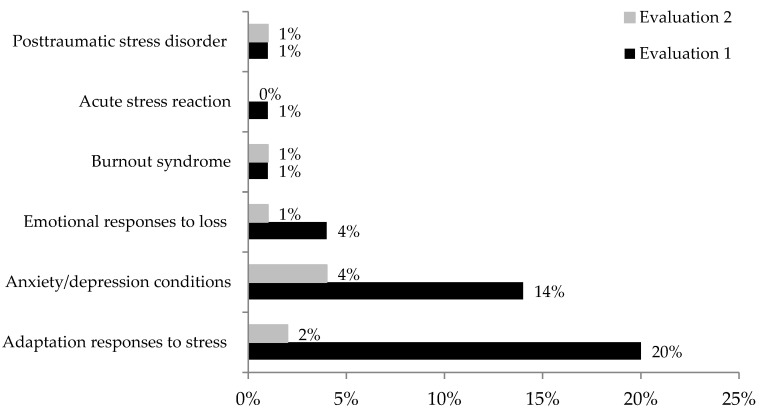
Frequency of adverse emotional reactions of 108 physicians during the two evaluations of the study (with three months in between).

**Figure 2 ijerph-20-03989-f002:**
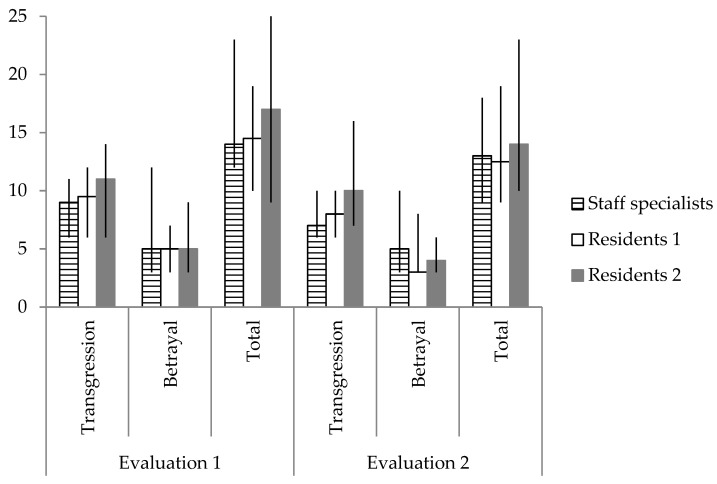
Median and interquartile range (Q1–Q3) of the total score and sub-scores on the Moral Injury Event Scale of 108 physicians in the two evaluations of the study, with three months in between. Residents 1 = residents since wave 1 = Residents 2 = residents since wave 3.

**Figure 3 ijerph-20-03989-f003:**
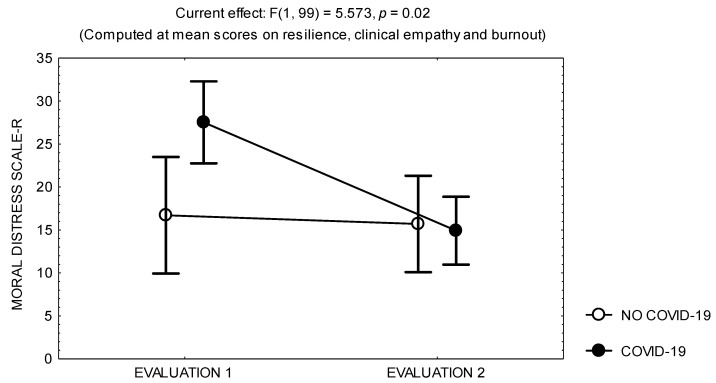
Mean and standard error of the mean of the scores on The Moral Distress Scale-R of 108 physicians in the two evaluations, according to the history of sick leave due to COVID-19.

**Table 1 ijerph-20-03989-t001:** Summary of the total scores on the assessment instruments of 108 participants (62 women/46 men) by sex, during the two evaluations. Comparisons between evaluations were performed using Wilcoxon matched pairs test.

Instruments	Evaluation 1 Median (Q1–Q3)	Evaluation 2 Median (Q1–Q3)	p (Z)
Hospital Anxiety and Depression Scale			
- Anxiety score (all)	6 (2.5–9)	-	-
Women	5 (2–9)	-	-
Men	6 (3–9)	-	-
- Depression score (all)	3 (1–4)	-	-
Women	3 (1–4)	-	-
Men	3 (1–4)	-	-
- Total score (all)	8 (4–13)	-	-
Women	8.5 (4–13)	-	-
Men	8 (4–13)	-	-
Resilience Scale (all)	78.5 (71–87)	-	-
Women	78 (72–87)	-	-
Men	78.5 (69–88)	-	-
Pittsburg Sleep Quality Index (all)	8.5 (6–11)	7 (5–10)	0.06 (1.851)
Women	9 (7–11)	7 (6–10)	0.003 (2.954)
Men	7 (5–9)	7 (5–9)	0.72 (0.355)
State-Trait Anxiety Inventory (all)	4 (2–6.5)	3 (1–6)	0.08 (1.748)
Women	4 (2–7)	4 (2–6)	0.48 (0.697)
Men	4 (2–6)	2 (1–5)	0.06 (1.827)
Burnout Measure (all)	2.5 (1.8–3.2)	2.2 (1.8–2.9)	0.003 (2.895)
Women	2.6 (1.9–3.4)	2.4 (2.0–3.1)	0.08 (1.720)
Men	2.4 (1.8–3.0)	2.1 (1.6–2.6)	0.01 (2.385)
Dissociative Experiences Scale (all)	3.9 (1.4–8.2)	2.6 (0.7–5.5)	0.003 (2.878)
Women	4.4 (1.0–7.8)	2.8 (0.7–5.0)	0.01(2.345)
Men	3.9 (1.7–10)	2.5 (0.3–5.7)	0.12 (1.552)
Depersonalization/derealization (all)	7 (2–14)	5 (1.5–10)	0.001(3.232) *
Women	6.5 (2–14)	5.5 (3–10)	0.09 (1.666)
Men	7 (3–14)	4 (1–9)	0.003 (2.949)
Orientation to Life Questionnaire (all)	68 (60.5–77.5)	72.5 (63.5–80)	0.007 (2.666)
Women	67 (71–77)	72 (62–80)	0.03 (2.142)
Men	68.5 (60–78)	73 (66–81)	0.10 (1.642)
Jefferson Scale of Empathy (all)	116 (106–127)	116 (105–127)	0.79 (0.254)
Women	119 (110–128)	118 (108–128)	.0.53 (0.627)
Men	111 (109–122)	114 (96–123)	1 (0)
Moral Sensitivity Questionnaire (all)	38.5 (34–43)	39 (35–41)	0.42 (0.792)
Women	39 (34–43)	39 (34–41)	0.43 (0.787)
Men	37.5 (35–42)	38.5 (35–41)	0.68 (0.402)
Items of Moral Distress Scale-R (all)	16 (8–35.3)	11.5 (3.0–23.5)	0.0004(3.508) *
Women	24.5 (10–40)	13 (7–27)	0.004 (2.828)
Men	13.5 (6–24)	9 (2–16)	0.03 (2.167)
Additional COVID-19 situations (all)	12 (4–18.5)	6 (1.5–12.5)	<0.0001(4.910) *
Women	13.5 (5–21)	7 (2–14)	<0.0001 (4.637) *
Men	8 (2–15)	4 (1–12)	0.03 (2.168)
Moral Injury Event Scale (all)	15 (10–20)	13 (9–19)	0.15 (1.412)
Women	15 (11–21)	13.5 (9–19)	0.55 (0.596)
Men	14 (10–19)	12.5 (9–21)	0.4 (.840)

* Significant after Holm-Bonferroni Correction [59].

**Table 2 ijerph-20-03989-t002:** Summary of the total scores on the assessment instruments of the 108 participants during the two evaluations, according to the experience group. Comparisons among groups were performed by Kruskal–Wallis ANOVA.

Instruments	Staff Physicians	Residents Since Wave 1	Residents Since Wave 3	p (H)
	Median (Q1–Q3)	Median (Q1–Q3)	Median (Q1–Q3)	
Hospital Anxiety and Depression Scale				
- Anxiety score	3 (1–8)	6 (3–10)	6 (2–9)	0.27 (2.567)
- Depression score	2 (0–3)	3 (2–5)	2 (1–5)	0.28 (2.507)
- Total score	6 (3–10)	10 (5–16)	8 (4–13)	0.18 (3.358)
Resilience Scale	87 (81–90)	76 (67–83)	80 (73–85)	0.08 (5.004)
Pittsburg Sleep Quality Index				
Evaluation 1	9 (4–11)	9 (6–11)	7 (5–9)	0.25 (2.724)
Evaluation 2	7 (6–9)	7 (5–10)	7 (5–11)	0.16 (3.620)
State-Trait Anxiety Inventory				
Evaluation 1	3 (1–5)	4 (2–7)	4 (3–6)	0.24 (2.822)
Evaluation 2	2 (1–4)	4 (2–7)	4 (1–6)	0.66 (0.828)
Burnout Measure				
Evaluation 1	2.3 (1.7–3.0)	2.6 (1.9–3.2)	2.6 (2–3)	0.29 (2.43)
Evaluation 2	1.9 (1.5–2.6)	2.4 (2.0–3.2)	2.3 (1.9–2.6)	0.10 (4.459)
Dissociative Experiences Scale				
Evaluation 1	2.5 (0.3–6.4)	4.6 (2.1–8.2)	3.9 (1.8–10.3)	0.19 (3.229)
Evaluation 2	2.5 (0.3–5.0)	3.0 (0.7–5.0)	2.1 (0.4–10.3)	0.90 (0.200)
Depersonalization/derealization				
Evaluation 1	2 (1–10)	8.5 (3–14)	7 (2–16)	0.07 (5.241)
Evaluation 2	3 (0–6)	6 (2–11)	6 (3–10)	0.18 (3.416)
Orientation to Life Questionnaire				
Evaluation 1	77 (67–82)	66 (57–72)	68 (61–78)	0.10 (4.518)
Evaluation 2	80 (72–86)	71 (59–79)	71 (65–79)	0.17 (3.500)
Jefferson Scale of Empathy				
Evaluation 1	120 (107–128)	115 (110–125)	118 (100–127)	0.77 (0.504)
Evaluation 2	118 (106–122)	115 (107–128)	115 (91–129)	0.93 (0.145)
Moral Sensitivity Questionnaire				
Evaluation 1	39 (34–47)	39 (35–43)	38 (33–43)	0.69 (0.74)
Evaluation 2	39 (34–41)	39 (36–41)	39 (32–40)	0.39 (1.840)
Items of Moral Distress Scale-R				
Evaluation 1	22 (5–41)	16 (10–32)	15/8–33)	0.85 (0.316)
Evaluation 2	10 (1–18)	13 (5–30)	10 (2–17)	0.52 (1.261)
Additional COVID-19 situations				
Evaluation 1	10 (3–18)	12 (5–18)	14 (4–22)	0.58 (1.068)
Evaluation 2	5 (0–9)	7 (2–16)	3 (1–12)	0.65 (0.848)
Moral Injury Event Scale				
Evaluation 1	14 (12–23)	15 (10–19)	17 (9–25)	0.93 (0.125)
Evaluation 2	13 (9–18)	13 (9–19)	14 (10–23)	0.23 (2.931)

**Table 3 ijerph-20-03989-t003:** Summary of responses to the moral distress items on COVID-19 situations of 108 physicians according to the experience group during the two evaluations. Comparisons among groups were performed by Kruskal Wallis ANOVA.

Items	Staff Physicians	Residents Since Wave 1	Residents Since Wave 3	p (H)
	Median (Q1–Q3)	Median (Q1–Q3)	Median (Q1–Q3)	
Caring for patients who must experience hospitalization without family presence				
Evaluation1	0 (0–4)	0.5 (0–4)	1 (0–6)	0.83 (0.349)
Evaluation 2	0 (0–1)	0.5 (0–2)	0 (0–1)	0.45 (1.555)
Caring for patients who die during a hospitalization without family and/or clergy present				
Evaluation 1	1 (0–4)	1.5 (0–4)	2 (0–4)	0.19 (3.273)
Evaluation 2	0 (0–2)	1.5 (0–4)	0 (0–2)	0.34 (2.113)
Being assigned/floated to a new unit, requiring unfamiliar skills or procedures				
Evaluation 1	0 (0–1)	3 (1–6)	4 (1–6)	0.0007 (9.880) *
Evaluation 2	0 (0–0)	3 (0–4)	2 (0–2)	0.11 (4.327)
Caring for patients who present transmission risk to your family/household				
Evaluation 1	6 (1–9)	2 (0–8)	4 (1–9)	0.84 (0.339)
Evaluation 2	1 (0–6)	2 (0–6)	1 (0–6)	0.87 (0.264)

* Significant after Holm–Bonferroni Correction [59].

**Table 4 ijerph-20-03989-t004:** Goodman–Kruskal gamma (G) correlation coefficients between moral distress and moral injury scores of 108 physicians, between evaluations 1 and 2 and within each evaluation.

Instrument				
Between Evaluations				
	Gamma	p (Z)		
Items of the Moral Distress Scale-R	0.224	0.0008(3.339)	-	-
Moral Distress COVID-19 situations	0.402	<0.0001 (5.906)	-	-
Moral Injury Event Scale Transgressions	0.348	<0.0001 (4.634)	-	-
Moral Injury Event Scale Betrayal.	0.245	0.002 (3.031)	-	-
Moral Injury Event Scale Total score	0.273	0.0001 (3.843)	-	-
Within each evaluation	Evaluation 1		Evaluation 2	
	Gamma	p (Z)	Gamma	p (Z)
Items of the Moral Distress Scale-R and				
- Transgressions on the Moral Injury Event Scale	0.261	0.0002 (3.678)	0.345	<0.0001 (4.767)
- Betrayal on the Moral Injury Event Scale.	0.364	<0.0001 (5.057)	0.230	0.002 (3.000)
- Total score on the Moral Injury Event Scale	0.361	<0.0001 (5.298)	0.327	<0.0001 (4.666)
COVID-19 situations and				
- Transgressions on the Moral Injury Event Scale	0.320	<0.0001 (4.499)	0.292	<0.0001 (3.988)
- Betrayal on the Moral Injury Event Scale.	0.353	<0.0001 (4.890)	0.319	<0.0001 (4.145)
- Total score on the Moral Injury Event Scale	0.363	<0.0001 (5.312)	0.344	<0.0001 (4.869)

**Table 5 ijerph-20-03989-t005:** Results of the multivariate analysis of covariance on the Moral Distress Scale -R score of 108 physicians between the two evaluations and within each evaluation.

Items of the Moral Distress Scale-R				
Within Each Evaluation	Evaluation 1		Evaluation 2	
	F	p	F	p
Intercept	4.846	0.029 *	0.779	0.379
Resilience Scale	0.439	0.508	6.055	0.015 *
Jefferson Scale of Empathy	6.869	0.010 *	4.290	0.040 *
Burnout Measure	18.952	<0.0001 *	1.66	0.199
COVID-19 sick leave	7.067	0.009 *	0.013	0.906
Repeated measure	6.113	0.015 *	-	-
Resilience Scale	5.200	0.024 *	-	-
Jefferson Scale of Empathy	0.577	0.449	-	-
Burnout Measure	7.648	0.006 *	-	-
COVID-19 sick leave	4.654	0.033 *	-	-

* Statistical significance ≤0.05.

**Table 6 ijerph-20-03989-t006:** Results of the multivariable analysis on the score and sub-scores on the Moral Injury Event Scale of 108 physicians, in the two evaluations.

Moral Injury Event Scale	Evaluation 1		Evaluation 2	
Total Score	Estimate (95% C.I.)	p (Wald Statistic)	Estimate (95% C.I.)	p (Wald Statistic)
Intercept	3.170 (2.610–3.730)	<0.0001 (123.01)	2.960 (2.399–3.522)	<0.0001 (106.79)
Orientation to Life Question.	−0.009 (−0.016–−0.002)	0.012 (6.258)	−0.005 (−0.012–0.002)	0.144 (2.134)
State-Trait Anxiety Inventory	−0.033 (−0.056–−0.009)	0.006 (7.420)	−0.035 (−0.060–−0.010)	0.005 (7.697)
Dissociative Experiences Scale	0.019 (0.007–0.031)	0.002 (9.026)	0.011 (0.001–0.021)	0.028 (4.803)
Moral Distress Scale-R	0.005 (0.001–0.010)	0.019 (5.444)	0.007 (0.002–0.013)	0.011 (6.430)
COVID-19 situations	0.012 (0.005–0.019)	0.0005 (12.123)	0.016 (0.006–0.026)	0.001 (10.617)
GROUP I (^residents 1/staff)	−0.055 (−0.151–0.041)	0.260 (1.268)	−0.053 (−0.148–0.042)	0.272 (1.203)
GROUP II (^residents 2/staff)	−0.044 (−0.151–0.063)	0.416 (0.660)	0.038 (−0.066–0.141)	0.474 (0.512)
Sick leave due to COVID-19	0.022 (−0.053–0.097)	0.563 (0.334)	0.099 (0.029–0.169)	0.005 (7.707)
GROUP I* COVID-19 sick leave	0.051 (−0.042–0.144)	0.285 (1.141)	0.074 (−0.015–0.163)	0.102 (2.672)
GROUP II* COVID-19 sick leave	−0.128 (−0.235–−0.021)	0.019 (5.486)	−0.088 (−0.192–0.017)	0.099 (2.718)
Transgression score				
Intercept	2.322 (1.645–3.000)	<0.0001 (3.000)	2.393 (1.773–3.014)	<0.0001 (57.093)
Orientation to Life Question.	−0.003 (−0.011–0.005)	0.49 (0.475)	−0.003 (−0.011–0.004)	0.407 (0.686)
State-Trait Anxiety Inventory	−0.028 (−0.056–0.000)	0.053 (3.740)	−0.034 (−0.062–0.007)	0.15 (5.899)
Dissociative Experiences Scale	0.013 (−0.002–0.028)	0.095 (2.788)	0.006 (−0.005–0.017)	0.273 (1.198)
Moral Distress Scale-R	0.004 (−0.001–0.010)	0.127 (2.326)	0.010 (0.004–0.016)	0.001 (10.439)
COVID-19 situations	0.014 (0.005–0.022)	0.001 (10.432)	0.012 (0.001–0.023)	0.029 (4.731)
GROUP I (^residents 1/staff)	−0.006 (−0.123–0.110)	0.913 (0.012)	−0.052 (−0.157–−0.053)	0.331 (0.944)
GROUP II (^residents 2/staff)	0.034 (−0.095–0.164)	0.604 (0.269)	0.127 (0.012–0.241)	0.030 (4.706)
Sick leave due to COVID-19	0.025 (−0.066–0.116)	0.587 (0.294)	0.114 (0.036–0.191)	0.004 (8.279)
GROUP I* COVID-19 sick leave	0.051 (−0.062–0.163)	0.378 (0.776)	0.074 (−0.024–0.173)	0.139 (2.184)
GROUP II* COVID-19 sick leave	−0.154 (−0.283–−0.024)	0.019 (5.417)	−0.109 (−0.224–0.006)	0.063 (3.444)
Betrayal score				
Intercept	2.700 (2.026–3.375)	<0.0001 (61.572)	2.166 (1.361–2.971)	<0.0001 (27.81)
Orientation to Life Question.	−0.017 (−0.026–−0.009)	<0.0001 (16.640)	−0.009 (−0.018–0.001)	0.079 (3.085)
State-Trait Anxiety Inventory	−0.038 (−0.067–−0.010)	0.007 (7.041)	−0.037 (−0.073–−0.002)	0.040 (4.193)
Dissociative Experiences Scale	0.029 (0.015–0.044)	0.0001 (15.073)	0.020 (0.006–0.034)	0.004 (8.008)
Moral Distress Scale-R	0.005 (0.000–0.010)	0.07 (3.279)	0.001 (−0.007–0.009)	0.872 (0.026)
COVID-19 situations	0.013 (0.005–0.021)	0.002 (9.322)	0.024 (0.010–0.038)	0.0006 (11.752)
GROUP I (^residents 1/staff)	−0.124 (−0.239–−0.008)	0.036 (4.380)	−0.037 (−0.173–0.099)	0.591 (0.288)
GROUP II (^residents 2/staff)	−0.177 (−0.306–−0.048)	0.0071 (7.232)	−0.140 (−0.288–0.009)	0.065 (3.391)
Sick leave due to COVID-19	0.015 (−0.075–0.106)	0.738 (0.112)	0.077 (−0.023–0.177)	0.132 (2.265)
GROUP I* COVID-19 sick leave	0.051 (−0.062–0.163)	0.376 (0.781)	0.062 (−0.066–0.190)	0.543 (0.904)
GROUP II* COVID-19 sick leave	−0.086 (−0.215–0.043)	0.189 (1.718)	−0.044 (−0.194–0.105)	0.563 (0.334)

* Interaction between variables. ^ R1 are residents who were at the forefront of the medical care of patients with COVID-19 since the first wave of contagions.R2 are residents since the third wave of contagions.

## Data Availability

The data are contained within the article. The datasets are available from the corresponding author upon reasonable request.

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
