# Peer review of "Physicians’ Distress Related to Moral Issues and Mental Health In-Between Two Late Waves of COVID-19 Contagions"

_ijerph, 2023, doi:10.3390/ijerph20053989_

Round 1
Reviewer 1 Report
The manuscript includes a very large battery of psychometric instruments, but they need to be ordered before being analyzed.
What was the outcome variable and what were the covariates?
A multivariate analysis is needed to define the outcome variable (moral distress and moral injury).
Finally, all the results must be adjusted for age, sex and seniority level of the physicians.
The sample is too low.
The manuscript needs further proofreading before moving on to a second round of review.
Author Response
The manuscript includes a very large battery of psychometric instruments, but they need to be ordered before being analyzed.
- We thank the reviewer for the comment; the description of all the Methods section has been edited for clarity.
What was the outcome variable and what were the covariates?
-We thank the reviewer for the valuable comment. The description of the aim of the study (at the end of the introduction) has been edited to allow the reader to identify that we assessed distress related to:
- two outcomes: 1. mental health (anxiety/ depression / burnout/ stress reactions) and 2. moral issues (distress/ injury)
- while taking into account (as cofactors) the in-hospital experience, the history of sick leave due to COVID-19, the quality of sleep, resilience, clinical empathy, sense of coherence and moral sensitivity.
A multivariate analysis is needed to define the outcome variable (moral distress and moral injury).
- Multivariate analysis on moral distress and multivariable analysis on moral injury are described in the Results section.
Finally, all the results must be adjusted for age, sex and seniority level of the physicians.
- We have edited the description of the results and the discussion to emphasise that the analyses were performed taking into account age, sex and seniority (in-hospital experience).
The sample is too low.
- The limitations paragraph was edited to include a comment on the small sample size. In the same paragraph we emphasise the need of replication and further studies. However, within this limitation, the study was designed and performed controlling for occupation and working environment, just in frontline physicians from a single centre, with almost 80% participation; to overcome the limitation, we included confidence intervals when required and, before asny statistical analysis, we carefully observed data distribution, in addition most of the results reported were highly significant,.
The manuscript needs further proofreading before moving on to a second round of review.
- We thank the reviewer for the comment. The manuscript has been revised and extensively edited for clarity and language.
Reviewer 2 Report
First of all, I would like to thank you for the opportunity to read your interesting paper entitled “Physicians’ distress related to moral issues and mental health in between two late waves of COVID-19 contagions.” This study was designed to assess mental health related to moral distress and moral injury, in between two late waves of COVID-19 contagions, in physicians who were at the forefront of care of patients with COVID-19 during the first two years of the pandemic, in the same institutional context, according to their in-hospital experience, and taking into account their adverse psychological reactions (anxiety/depression/burnout/stress reactions), sick leave due to COVID-19, quality of sleep, moral sensitivity, clinical empathy, resilience, and sense of coherence. The results suggest that strengthening resilience and a sense of coherence, as well as personal health protection to prevent infection, may be helpful to prevent persistent mental damage after exposure to a sanitary crisis, such as COVID-19 waves of contagions.
However, some concerns in your study need to be addressed. I hope my comments below can be helpful for you as you improve this manuscript to deliver its full potential.
Please justify the need for the study in the introduction. The authors should focus more on addressing what we already know about the topic before bringing in a gap considering what the paper tries to fill in. This would make it clear to the reader why it is crucial to address the shortcomings in the literature.
How did you select your sample?
Please add the Appendix B (Instrument) in the measure section as main part of the document.
Please explain more about the validity of the items. How did you know that the items were validated? We want to know the details.
It is important to run the convergent and discriminant validity.
CFA analysis should be performed.
It is important to perform post hoc tests for the justification of the actual sample?
The theoretical and practical implications part is missing in the discussion section.
I suggest that the authors incorporate recent and more context-related articles related to variables.
The conclusion is a mere synthesis of the research findings. It is underdeveloped and falls short in stressing and arguing the original contribution of this research. It should be revised, trying to emphasize the value of this research.
Again, I enjoyed reading your paper and hope my comments can be helpful to you as you improve your manuscript.
Author Response
First of all, I would like to thank you for the opportunity to read your interesting paper entitled “Physicians’ distress related to moral issues and mental health in between two late waves of COVID-19 contagions.” This study was designed to assess mental health related to moral distress and moral injury, in between two late waves of COVID-19 contagions, in physicians who were at the forefront of care of patients with COVID-19 during the first two years of the pandemic, in the same institutional context, according to their in-hospital experience, and taking into account their adverse psychological reactions (anxiety/depression/burnout/stress reactions), sick leave due to COVID-19, quality of sleep, moral sensitivity, clinical empathy, resilience, and sense of coherence. The results suggest that strengthening resilience and a sense of coherence, as well as personal health protection to prevent infection, may be helpful to prevent persistent mental damage after exposure to a sanitary crisis, such as COVID-19 waves of contagions.
- We thank the reviewer for the kind comment.
However, some concerns in your study need to be addressed. I hope my comments below can be helpful for you as you improve this manuscript to deliver its full potential.Please justify the need for the study in the introduction. The authors should focus more on addressing what we already know about the topic before bringing in a gap considering what the paper tries to fill in. This would make it clear to the reader why it is crucial to address the shortcomings in the literature.
-The introduction has been edited accordingly.
How did you select your sample?
- Since the study was designed to control for occupation and the working environment, we decided to invite all potential candidates in a single centre, as long as “ they were physicians working at the Departments of Emergency Care, Internal Medicine, Respiratory Support, Anesthesia and Intensive Care, who have provided medical care to patients with COVID-19 during any of the first four waves of contagions (March 2020 to March 2022).”
- The Section 2.2 has been edited for clarity.
Please add the Appendix B (Instrument) in the measure section as main part of the document.
- Appendix B is now included in the Methods section of the main document.
Please explain more about the validity of the items. How did you know that the items were validated? We want to know the details. It is important to run the convergent and discriminant validity. CFA analysis should be performed.
- The description of the instruments (Section 2.4) has been edited to describe the original validation of the items and instrument to assess moral distress and moral injury respectively; including the results on factor analysis and internal consistency.
It is important to perform post hoc tests for the justification of the actual sample?
- We agree with the reviewer that power analysis plays a key role in designing prospective studies. However, evidence has shown that when power analysis is used for outcomes already observed, it is analytically misleading, and do not provide sensible results (Zhang 2019); while confidence intervals may better inform about the possibility of an inadequate sample size for the analysis (Levine 2012).
- According to the design of this study, we invited all the available candidates, with a positive response from 78.4% of them, and a single elimination due to incomplete responses. We provide confidence intervals when required.
- Levine M., Ensom MHH. Post Hoc Power Analysis: An Idea Whose Time Has Passed? Pharmacotherapy 2001; 21 (4): 405-409. DOI: 10.1592/phco.21.5.405.34503
-Zhang Y, Hedo R, Rivera A, Rull R, Richardson S, Tu XM. Post hoc power analysis: is it an informative and meaningful analysis? Gen Psychiatr. 2019 Aug 8;32(4):e100069. doi: 10.1136/gpsych-2019-100069
The theoretical and practical implications part is missing in the discussion section. I suggest that the authors incorporate recent and more context-related articles related to variables.
- We thank the reviewer for the comments, we have edited the introduction and the discussion accordingly, while incorporating more references.
- However, we tried to keep the discussion as simple as possible, to make the document appealing to a broad readership.
The conclusion is a mere synthesis of the research findings. It is underdeveloped and falls short in stressing and arguing the original contribution of this research. It should be revised, trying to emphasize the value of this research.
- We thank the reviewer. The conclusion has been edited accordingly.
Again, I enjoyed reading your paper and hope my comments can be helpful to you as you improve your manuscript.
- The comments were very helpful and we truly appreciate them.
Round 2
Reviewer 2 Report
Dear Authors, I carefully re-evaluated your paper, finding it substantially improved with respect to the version. The revised version is much better organized and has higher scientific quality. Therefore, I recommended it for publication. Thank you